# Prematurity and Epigenetic Regulation of *SLC6A4*: Longitudinal Insights from Birth to the First Month of Life

**DOI:** 10.3390/biomedicines13112753

**Published:** 2025-11-11

**Authors:** Aline de Araújo Brasil, Leo Travassos Vieira Milone, Paulo Victor Barbosa Eleutério dos Santos, Stephanie Cristina Alves de Oliveira Saide, Vitor Barreto Paravidino, Georgia Chalfun, Letícia Santiago da Silva Ferreira, Mariana Berquó Carneiro Ferreira, Anna Beatriz Muniz Ferreira, Geovanna Barroso de Farias, Jaqueline Rodrigues Robaina, Mariana Barros Genuíno de Oliveira, Maria Clara de Magalhães-Barbosa, Arnaldo Prata-Barbosa, Antonio José Ledo Alves da Cunha

**Affiliations:** 1Department of Pediatrics, D’Or Institute for Research and Education (IDOR), Rio de Janeiro 22281-100, RJ, Brazil; aline.brasil@idor.org (A.d.A.B.); leo.milone@idor.org (L.T.V.M.); paulo.eleuterio@idor.org (P.V.B.E.d.S.); stephanie.saide@idor.org (S.C.A.d.O.S.); gechalfun@gmail.com (G.C.); leticia.silvaf@idor.org (L.S.d.S.F.); mariana.carneiro@idor.org (M.B.C.F.); anna.muniz@idor.org (A.B.M.F.); geovannab.farias@gmail.com (G.B.d.F.); jaqueline.robaina@idor.org (J.R.R.); mariana.genuino@idor.org (M.B.G.d.O.); mariaclara.magalhaes@idor.org (M.C.d.M.-B.); acunha@hucff.ufrj.br (A.J.L.A.d.C.); 2Department of Epidemiology, Institute of Social Medicine, State University of Rio (UERJ), Rio de Janeiro 20550-900, RJ, Brazil; vparavidino@gmail.com; 3National School of Public Health, Oswaldo Cruz Foundation, Rio de Janeiro 21040-900, RJ, Brazil; 4Department of Physical Education and Sports, Naval Academy, Brazilian Navy, Rio de Janeiro 20021-010, RJ, Brazil; 5Department of Neonatology, Maternity School, Federal University of Rio de Janeiro (UFRJ), Rio de Janeiro 22240-003, RJ, Brazil; 6Department of Pediatrics, School of Medicine, Federal University of Rio de Janeiro (UFRJ), Rio de Janeiro 21941-902, RJ, Brazil

**Keywords:** prematurity, epigenetics, serotonin transporter, *SLC6A4*, DNA methylation, neonatal stress, neurodevelopment

## Abstract

**Background/Objectives:** Prematurity is a significant global health concern, often associated with neurodevelopmental challenges. Solute Carrier Family 6 Member 4 (*SLC6A4*), the gene encoding the serotonin transporter, a key component in serotonin reuptake in the synaptic cleft, plays a key role in stress response and neurodevelopment. Epigenetic regulation of stress-related genes, such as *SLC6A4*, influences neonatal stress adaptation and developmental outcomes. This study aimed to quantify and compare DNA methylation levels at 13 CpG sites in the promoter region of the *SLC6A4* gene between preterm and term neonates at three time points. **Methods:** A cohort of 46 preterm infants and a cohort of 49 full-term infants were analyzed. Blood samples collected at birth (D0), the fifth day (D5), and the thirtieth day (D30) were used to analyze DNA methylation, using bisulfite conversion and pyrosequencing. **Results:** Significant differences in *SLC6A4* methylation were observed. At D0, CpGs 12 and 13 showed higher methylation in preterm infants. CpG 9 showed lower methylation in preterm infants at D5. Extremely preterm infants had the highest values of methylation at the three time points. Longitudinal mixed-effects analysis revealed distinct temporal patterns between groups. Total and site-specific methylation at CpGs 2, 8, and 9 increased over time in full-term infants, while methylation remained stable over time in very preterm and extremely preterm infants. **Conclusions:** This study reveals significant differences in *SLC6A4* methylation between very preterm, extremely preterm, and full-term infants, highlighting the impact of prematurity and early-life stress on the epigenome. These findings contribute to improving our understanding of the epigenetic mechanisms shaping neurodevelopment and stress adaptation in neonates.

## 1. Introduction

Prematurity is a significant global health challenge, with it being the main cause of death in children under five years of age [1]. Beyond increased mortality, it is associated with early-life morbidities like neurodevelopmental delays [2] and higher risks of psychiatric conditions, including autism, anxiety, depression, and attention-deficit/hyperactivity disorder (ADHD) [3,4]. Several risk factors may be implicated in these findings, including comorbidities such as brain injuries and bronchopulmonary dysplasia, as well as socioeconomic factors [5]. Additionally, very preterm infants in Neonatal Intensive Care Units (NICUs) face stressors like pain, maternal separation, noise, and invasive procedures, which can affect brain organization, hypothalamic–pituitary–adrenal (HPA) axis regulation, and cognitive outcomes [6,7,8,9,10,11], besides further influencing the epigenetic regulation of stress-response genes [12,13,14,15].

In neonatal care, several strategies have been implemented to minimize stress and its potential long-term effects, including pain relief interventions, skin-to-skin care (Kangaroo Mother Care), exposure to the human voice, and early communication approaches with infants in the NICU environment [16,17,18,19,20,21]. Although most of these approaches are supported primarily by animal research and a few human studies with relatively small sample sizes, they represent essential strategies to mitigate the adverse effects of early stress on infant development. Understanding the biological mechanisms underlying these effects, including epigenetic regulation of stress-related genes, may help strengthen the scientific basis for such interventions and guide future clinical strategies.

Serotonergic neurotransmission is a key modulator of responses to environmental stressors and is closely linked to central nervous system (CNS) development during the early years of life [22,23]. This pathway is primarily regulated by the reuptake of serotonin from the synaptic cleft via the serotonin transporter (5-HTT), located on the presynaptic neuron membrane. In humans, this transporter is encoded by the *SLC6A4* gene (Solute Carrier Family 6 Member 4), situated on chromosome 17q11.2 [24] (Figure 1). The *SLC6A4* gene exhibits structural variations, including the 5-HTTLPR polymorphism, which comprises short and long alleles. The short allele reduces serotonin reuptake by 50% compared to the long allele [24] and is associated with heightened stress sensitivity and emotional outcomes during infancy and childhood [25]. Beyond genetic influences, the transcriptional activity of *SLC6A4* is subject to epigenetic regulation [26,27], with studies linking early life experiences to changes in the gene’s methylation patterns [28,29].

DNA methylation regulates gene expression by blocking transcription factor binding and promoting heterochromatin formation, affecting nearly half of human genes [30]. Clinically, DNA methylation offers a dynamic tool for understanding complex diseases, diagnosing and predicting conditions, and identifying biomarkers, surpassing the static nature of polygenic risk scores with its adaptability for longitudinal disease evaluation [31,32,33]. The methylation levels at distinct CpG sites within the *SLC6A4* promoter region have been inversely associated with serotonin transporter expression and have been explored as an important target in neuropsychological outcomes such as depression, anxiety, and burnout [34,35,36], as well as being associated with stress response and emotional regulation during early life [37,38]. Thus, the *SLC6A4* gene is proposed to play a pivotal role as an epigenetic mediator, influencing how early adverse experiences shape the developmental trajectories of infants and children.

An increase in *SLC6A4* methylation in preterm infants at NICU discharge has been observed following exposure to high levels of pain-related stress during hospitalization [8]. However, no previous study has analyzed the methylation pattern of this gene across different time points within the first month of life in both preterm and term infants.

This study hypothesizes that preterm neonates exhibit distinct DNA methylation patterns in the *SLC6A4* promoter region compared to term neonates, driven by their exposure to early-life stressors such as those encountered in the NICU. These epigenetic differences might influence the regulation of serotonin transporter activity, shaping stress responses and neurodevelopmental outcomes. This research aims to quantify and compare the methylation levels of 13 CpG sites in the promoter region of the *SLC6A4* gene between preterm and term neonates at three time points in the first month of life.

## 2. Materials and Methods

### 2.1. Study Design, Study Population and Ethics

This is an observational, longitudinal study in two previously described cohorts, which were established to investigate the DNA methylation of specific genes [39]. The cohorts included 46 adequate-for-gestational-age preterm infants weighing ≤ 1500 g, consecutively admitted to the NICU of a university maternity hospital from April 2018 to May 2019, and 49 healthy full-term infants born in the same period at the same maternity hospital [32]. Infants classified as small for gestational age (SGA), large for gestational age (LGA), or with major congenital malformations suggestive of genetic syndromes (clinically suspected or laboratory-confirmed) were excluded. This study was approved by the ethics committees of the participating institutions, and informed consent was obtained from all parents or legal guardians of the infants. All data were anonymized.

### 2.2. Study Variables

Sociodemographic and clinical variables of the infants and pregnant women collected from medical records were available in the dataset. For the present study, the preterm group was further subdivided into “extremely preterm” (<28 weeks) and “very preterm” (28 to <32 weeks) categories, based on the World Health Organization classification (https://www.who.int/news-room/fact-sheets/detail/preterm-birth, accessed on 10 November 2025). Gestational age was determined via first-trimester ultrasound (USG). In cases where USG data were unavailable, gestational age was estimated using the date of the last menstrual period (LMP) and either the Capurro or Ballard method [40,41].

### 2.3. Biological Material Collection

The following previously collected, and adequately stored samples were used to *SLC6A4* gene methylation analysis: umbilical cord blood samples from preterm and full-term infants (D0); peripheral blood samples from the fifth day of life of both groups (D5); peripheral blood samples from the 30th day of life from preterm infants remaining in the Neonatal Intensive Care Unit (NICU) (D30). All blood samples were collected into tubes containing potassium EDTA (K2EDTA) as an anticoagulant. After collection, samples were stored at 4–8 °C for up to 48 h before being transported in thermal containers to the Epigenetics Laboratory. Upon arrival, samples were aliquoted, labelled, and stored at −80 °C until subsequent processing.

### 2.4. SLC6A4 Methylation Analysis

The region of the *SLC6A4* gene analyzed consisted of the CpG islands in the promoter region (chr17:28562750.28562958, Human hg19 Assembly), between −69 and −213, corresponding to 13 CpG sites adjacent to exon 1A. DNA was extracted from the collected blood using the DNeasy Blood and Tissue Kit (Qiagen, Hilden, Germany), following the manufacturer’s instructions. The concentration and purity of the genomic DNA (gDNA) were measured by spectrophotometry using the NanoDrop^TM^ 2000c spectrophotometer (ThermoFisher Scientific, Waltham, MA, USA). Subsequently, 1000 ng of purified gDNA were subjected to sodium bisulfite conversion using the EZ DNA Methylation™ Kit (Zymo Research, Tustin, CA, USA). Next, 2 µL of bisulfite-converted DNA was amplified by polymerase chain reaction (PCR) using the Veriti 96-Well Thermal Cycler (Applied Biosystems, Waltham, MA, USA) and the PyroMark PCR Kit (Qiagen, Hilden, Germany) in a 50 µL reaction volume. Amplification was confirmed by 3% agarose gel electrophoresis. Following amplification, 10 µL of DNA was immobilized on streptavidin-coated Sepharose beads and sequenced using the PyroMark Q48 Autoprep System (Qiagen, Hilden, Germany). Pyrosequencing primers were designed using PyroMark Assay Design 2.0 software (version 2.0), and methylation percentages were determined using PyroMark Q48 Autoprep Software (version 2.4.2) (both from Qiagen, Hilden, Germany) under standard quality control conditions. Human Methylated and Non-methylated DNA Set (Zymo Research, CA, USA) was used as positive and negative controls in methylation detection assays. All PCR and pyrosequencing primers, as well as the analyzed sequences, are detailed in Table 1.

### 2.5. Statistical Analysis

Maternal and neonatal sociodemographic and clinical characteristics of full-term and preterm infants were described and presented in tables. Continuous variables were reported as mean and standard deviation (SD) or median and interquartile range (IQR). Categorical variables were reported as percentages. The methylation percentages of each CpG and the total methylation of the *SLC6A4* gene (mean methylation of the 13 CpGs) of full-term, very preterm, and extremely preterm infants were reported as median and interquartile range (IQR) and compared using the Mann–Whitney U test. They were presented in tables and violin plots. Longitudinal analyses of methylation percentages were conducted for each CpG site individually and for total *SLC6A4* methylation using linear mixed-effects models with the PROC MIXED procedure in SAS OnDemand for Academics (version 9.4, 2023, SAS Institute Inc., Cary, NC, USA). We fitted parametric curves to the data, using an unstructured covariance matrix and including, in the model, the terms time, group, the interaction “time*group”, and a minimum set of confounding variables determined by a directed acyclic graph (DAG). When assessing differences in methylation between groups at each of the 13 CpGs, the Benjamini–Hochberg method was used for correction for multiple tests. However, since this is an exploratory study, a false discovery rate [FDR] of 10% (*q* < 0.10) was used.

## 3. Results

### 3.1. Epidemiological and Clinical Characteristics of the Study Population

The main epidemiological and clinical characteristics of the study population are summarized in Table 2. Maternal education exceeded 12 years in more than half of the preterm and full-term groups. In the preterm group, 28.3% of families had an income below the national minimum wage, compared to 8.2% in the term group. Smoking and alcohol consumption were low in both groups. Differences in the number of prenatal consultations, birth-related characteristics, and anthropometric measurements between the groups reflected gestational age, while the sex distribution remained balanced.

### 3.2. Methylation Percentage at Different Time Points

Figure 2 displays violin plots representing the quartile distribution of total methylation of the *SLC6A4* gene. Preterm infants exhibited significantly higher median of total methylation at D0 compared to full-term infants (2.55% vs. 2.18%, *p* = 0.004; Figure 2A). No significant differences were found between full-term and preterm infants at D5 (Figure 2B). When preterm infants were stratified into very preterm and extremely preterm, both showed significantly higher values compared to full-term infants at D0 (2.55% vs. 2.18%, *p* = 0.016 and 2.97% vs. 2.18%, *p* = 0.022, respectively; Figure 2D). At D5, a significant difference was observed only between full-terms and extremely preterm infants (2.26% vs. 3.34%, *p* = 0.017; Figure 2E). No difference between very preterm and extremely preterm infants was observed at D30 (Figure 2F). Extremely preterm infants had the highest values of total methylation at all three time points.

To further investigate site-specific patterns, Table 3 and Figure 3 present the median and interquartile ranges (IQRs) of methylation percentages at each CpG site, comparing full-term and preterm infants, at the three time points. Considering the methylation levels of all CpGs in both groups and at all time points, median values varied from 0.81% to 2.95% at CpGs 1 to 11, while CpGs 12 and 13 had the highest median values, varying from 3.57 to 7.64% (Table 3). At D0, differences were observed at CpG 12 and 13, which were significantly more methylated in preterm infants, compared to full-term infants (*p* = 0.007 for both CpGs) (Figure 3C,E). At D5, only CpG 9 differed between groups, being hypomethylated in preterm compared to full-term infants (*p* = 0.013) (Figure 3B).

To better capture differences within the preterm group, Table 4 and Figure 4 display and compare the median and IQRs of methylation percentages at each CpG site in full-term, very preterm, and extremely preterm infants, at the same time points. At D0, differences were observed at CpGs 9, 10, 12, and 13. CpGs 12 and 13 showed significantly higher methylation in very preterm (*p* = 0.007 for both CpGs) and extremely preterm infants (*p* = 0.020 for both CpGs) compared to full-term infants (Figure 4E and Figure 4G, respectively). CpG 10 had higher methylation in extremely preterm (*p* = 0.091) compared to very preterm infants (Figure 4C), while CpG 9 had lower methylation in extremely preterm (*p* = 0.087) compared to full-term infants (Figure 4A). At D5, differences persisted at CpGs 9, 12, and 13. CpG 12 and CpG 13 remained significantly higher in extremely preterm infants (*p* = 0.013 for both CpGs), compared to full-term infants (Figure 4F and Figure 4H, respectively), while CpG 9 showed lower methylation in both very preterm (*p* = 0.013) and extremely preterm infants (*p* = 0.074) compared to full-term infants (Figure 4B). At D30, no difference was observed between very preterm and extremely preterm infants. Extremely preterm infants had the highest median values of total methylation (ranging from 2.97% to 3.34%) and site-specific methylation at CpGs 12 and 13 (ranging from 8.24% to 10.37%) across the three time points.

### 3.3. Longitudinal Dynamics of SLC6A4 Methylation in Preterm Infants

The minimum adjustment set determined by the DAG included the following variables: antenatal corticosteroids, gestational complications, maternal underlying diseases, twinning, neonatal bacterial infections, and congenital infections (Figure 5).

The longitudinal analysis using mixed-effect models revealed distinct temporal trajectories of *SLC6A4* methylation in the different groups (Table 5 and Figure 6). Full-term term infants exhibited a significant increase in total methylation over time (*p* = 0.0001), while very preterm and extremely preterm infants showed stable total methylation levels, with no significant changes throughout the first month of life (*p* = 0.3806 and *p* = 0.9583, respectively) (Figure 6A). At the site-specific level, full-term infants showed consistent increases at CpG2 (*p* = 0.0195), CpG8 (*p* = 0.0828), and CpG9 (*p* = 0.0195) (Figure 6B–D, respectively), while the other CpG sites did not exhibit statistically significant changes. In very preterm and extremely preterm infants, methylation remained stable at all CpG sites.

## 4. Discussion

The study investigated DNA methylation patterns in the *SLC6A4* gene in extremely preterm, very preterm, and full-term infants at birth, on day 5, and on day 30, revealing some differences among them. Preterm infants showed higher total methylation and some site-specific CpG differences at birth compared to full-term infants, with extremely preterm infants displaying the highest levels of methylation. The longitudinal mixed-effects analysis revealed distinct temporal trajectories of *SLC6A4* methylation in the different groups. Full-term infants exhibited a significant increase in total methylation during the first month of life, particularly at CpG2, CpG8, and CpG9. In contrast, very preterm and extremely preterm infants showed largely stable levels of methylation. Together, these findings suggest that gestational maturity influences *SLC6A4* methylation patterns, with full-term infants exhibiting greater epigenetic plasticity over time, whereas preterm infants display more restricted and stable temporal signatures, which may limit their adaptive capacity and increase their susceptibility to adverse outcomes.

The expression of the *SLC6A4* gene, which encodes the serotonin transporter, is influenced by both the genetic polymorphism (5-HTTLPR) and the methylation state of its promoter region, which is modulated by environmental factors, especially stressful situations. Studies associate the variants “short” and “long_G”, and the hypermethylation with low gene transcription and, conversely, the “long_A” variant and the hypomethylation with higher transcription [42,43,44,45,46]. The result is a variation in serotonin levels in the synaptic cleft of serotonergic neurons, which may explain human behaviors associated with anxiety and depression [47,48]. In preterm infants, who are particularly vulnerable to early stress, understanding *SLC6A4* methylation is essential [49,50]. Recent studies suggest that premature birth, in itself, is not necessarily associated with changes in *SLC6A4* gene methylation at birth [8,49]. However, these findings are still incipient, and this study aims to contribute to a better understanding of this research area.

The present study underscores significant differences in the methylation levels of the *SLC6A4* gene between preterm and full-term infants at specific CpG sites at three key time points (D0, D5, and D30). At birth (D0), preterm infants exhibited higher methylation levels at CpG 12 and CpG 13 compared to full-term infants. Conversely, CpG 9 showed lower methylation in preterm infants compared to full-term counterparts on days 0 and 5. By D5, elevated methylation at CpG 13 persisted in preterm infants compared to full-term newborns. At D30, analysis focused solely on preterm groups, revealing no differences between very preterm and extremely preterm infants. Stratification further emphasized distinct methylation patterns among full-term, very preterm, and extremely preterm infants, highlighting dynamic epigenetic modifications across developmental stages.

A significant increase in *SLC6A4* gene methylation has been observed in preterm infants by Provenzi et al. (2015) after exposure to painful procedures in the NICU, which are known to induce DNA methylation in stress-related genes [8]. After observing no differences in the methylation of CpG sites in the promoter region of the *SLC6A4* gene between 32 full-term and 56 preterm infants at birth, the authors found that preterm infants exposed to high, but not low, levels of pain-related stress, such as skin puncture procedures, showed a significant increase in methylation at two specific sites (CpG 5 and CpG 6) at the time of NICU discharge. The authors suggested that prematurity per se is not associated with epigenetic changes in the *SLC6A4* gene, but the intensity of stress exposure plays a crucial role [8]. In the present study, we employed a numbering system identical to that used by Provenzi et al. (2015) to identify the 13 CpG sites, and our sample size was comparable to theirs [8]. However, our study identified differences between groups in total and site-specific methylation (CpGs 9, 10, 12, and 13) as early as D0. In contrast to those previous findings, our study suggests that prematurity per se is already associated with distinct and site-specific methylation patterns at birth.

Other studies support the hypothesis that epigenetic changes in stress-related genes induced by prenatal stress in infants may already be present at birth [51,52]. Investigating the relationship between stress from the COVID-19 pandemic during pregnancy, the methylation of the *SLC6A4*, and child development, Provenzi et al. (2021) involved 108 mother-infant pairs at delivery and revealed that prenatal stress was associated with increased methylation of *SLC6A4* in infants in the first 24h of life at seven CpG sites (i.e., 1, 2, 6, 8, 9, 10, and 12), which predicted temperament traits such as higher surgency at 3 months of age, without affecting the methylation in mothers [51]. Nazzari et al. (2022) expanded the research to 283 pairs, finding that mothers and infants exposed to lockdown in the first trimester exhibited lower methylation of both *SLC6A4* and the glucocorticoid receptor gene (*NR3C1*), suggesting that epigenetic regulation is more sensitive in the second and third trimesters due to increased cortisol and inflammatory cytokines [52]. Both studies highlight the vulnerability of pregnant women and their fetuses to prenatal stress. These findings reinforce that prematurity per se, as an indicator of prenatal stress, may be associated with distinct and specific methylation patterns at birth. Furthermore, they point to the methylation of *SLC6A4* and *NR3C1* as potential biomarkers for the effects of prenatal stress on child development.

Other genes have been studied in the same cohort of the present study, showing altered methylation patterns in preterm infants at birth. Methylation changes of *NR3C1*, a stress-related gene [39], and *LINE-1*, a global DNA methylation marker [53], were reported in full-term and preterm infants at the same three time points (D0, D5, and D30). At birth, preterm infants exhibited hypermethylation at three CpG sites of the *NR3C1* gene (CpG 12, 42, and 47), but hypomethylation predominated later [39]. Methylation of the LINE-1 promoter, a global DNA methylation marker, was significantly reduced in preterm newborns at birth compared to full-term infants, with this difference decreasing over time. Both studies, including the present one, suggest that prenatal stress, intrinsic to prematurity, affects the neonatal epigenome by altering methylation patterns. Additionally, they highlight the role of postnatal factors in modulating these epigenetic changes in preterm infants. The environment of the NICU may play a key role in the potential normalization of the epigenetic profile.

The longitudinal mixed-effects analysis used in the present study allowed us to assess changes over time while accounting for both group-level effects and individual variability. Using this analysis, we found that only full-term infants exhibited a significant increase in *SLC6A4* total methylation during the first month of life. In contrast, total methylation remained stable in very preterm and extremely preterm infants, with no significant changes. Apparently, these results also contrast with previous findings mentioned above, reporting an increase in *SLC6A4* gene methylation associated with high levels of pain-related stress in preterm infants [8]. Dokkum et al. (2023) also suggest a potential relationship between neonatal stress exposure and *SLC6A4* methylation, as they found moderately high correlation coefficients between gene methylation and scores on the Neonatal Infant Stressor Scale (NISS) at NICU discharge, although without statistical significance [54]. Although we did not assess stress levels, we can assume that extremely preterm infants experience greater intensity and duration of painful stimuli than very preterm infants. Still, methylation was stable in both groups of preterm infants. On the other hand, converging with a potential relationship between neonatal stress exposure and *SLC6A4* methylation, extremely preterm infants had the highest values of total and some site-specific methylation across all three time points.

When individual CpG sites were examined in the longitudinal analysis of the present study, we observed significant changes over time only in full-term infants, with a significant increase in methylation at CpG2, CpG8, and CpG9. These findings suggest that the epigenetic response is not uniformly distributed across the promoter region but instead concentrated in specific CpG sites, which may be more sensitive to neonatal environmental conditions. Moreover, the results of this longitudinal analysis support the hypothesis that full-term birth is associated with greater epigenetic plasticity, reflected in more dynamic and targeted adjustments at regulatory regions, which may be critical for fine-tuning gene expression and optimal neurodevelopmental outcomes. In contrast, very preterm and extremely preterm infants did not show significant increases in methylation over time, suggesting a reduced capacity for epigenetic adaptation in the context of prematurity.

Provenzi et al. (2018) reported recurrent patterns of altered methylation of the *SLC6A4* gene after exposure to the neonatal ICU in a systematic review that consolidated findings in preterm infants. This review reported studies that identified specific effects at sites such as CpG2, 5, 6, 16, and 20 across different cohorts, but methodological heterogeneity (different CpG panels, tissues, and numbering) hinders direct synthesis. Furthermore, few studies specified the exact function of individual CpG sites in gene expression in relevant tissues (especially the brain) and in neurobehavioral clinical outcomes, particularly in preterm infants [50].

In functional model studies, in vitro methylation of a promoter fragment robustly reduces luciferase activity, suggesting a direct mechanism of transcriptional silencing by promoter methylation [55]. In human samples, Wang et al. (2012) identified a point association between CpG11–12 and reduced 5-HT levels in the orbitofrontal cortex, as measured by PET, supporting the idea that local methylation alterations correlate with in vivo serotonin measurements [55]. Other studies demonstrated a correlation between methylation and reduced levels of *SLC6A4* mRNA [56,57] and an association between methylation levels in multiple CpG sites in the *SLC6A4* promoter region with depressive symptoms, HPA axis reactivity, amygdala activation patterns, anxiety, and burnout [34,35,36,44,58]. However, the numbering/index of CpG is not standardized across these studies (each group usually reports the order within the fragment they analyze), which makes a direct correlation between specific sites difficult without mapping the exact genomic distributions.

In preterm infants, Chau et al. (2014) [59] stated that 7/10 CpGs were significantly more methylated in very preterm infants (VPT) of school age (~7 years) (CpG-1, 3–5, 8–10), while CpG-2 showed lower methylation, compared to full-term infants (FT). Methylation increased correlated with behavioral problems, and this effect interacted with neonatal pain exposure and COMT genotype. Although the 10 CpG sites analyzed by Chau et al. differ from those analyzed in this study, they are located within the same CpG island, situated in the promoter region of the *SLC6A4* gene, adjacent to exon 1a [59]. In our study, we did not investigate the *SLC6A4* polymorphism. The findings of Chau et al. highlight the complex interplay between epigenetics, genetic predisposition, and early stress in shaping neurodevelopmental and behavioral outcomes in preterm children, underscoring the importance of incorporating genotype assessment into epigenetic studies. In a recent longitudinal study, Provenzi et al. (2020) reported that CpG-specific methylation in the neonatal period (a pain-related increase measured at discharge) was a predictor of greater anger expression/greater emotional dysregulation at 4.5 years in preterm infants. Both CpG5 and CpG9 were significantly associated with reactive anger in response to induced stress [38]. As mentioned previously, we used the same numbering system as they do to identify the 13 CpG sites. Therefore, when possible, we recommend either referencing positions to the reference genome or including a comparative table with coordinates to facilitate comparison between studies.

Other studies associate the *SLC6A4* gene methylation with neurodevelopmental outcomes without specifying the CpG sites9. Higher *SLC6A4* methylation in very preterm infants at term-equivalent age was linked to reduced anterior temporal lobe volume, which was subsequently associated with suboptimal socio-emotional development at 12 months of corrected age [29]. Additionally, *SLC6A4* methylation levels have been associated with various outcomes, including anger responses to emotional stress at 4.5 years of age [38], behavioral problems in school-aged children [59], cortisol reactivity to stress [58], and hippocampal gray matter volume in adults [60,61]. These findings underscore the significance of the *SLC6A4* gene in neurodevelopmental and behavioral outcomes, highlighting the need for further research in this area. However, given the exploratory nature of the current studies, which sometimes yield contradictory results, more robust research is needed to enhance our understanding of the epigenetic mechanisms that shape neurodevelopment and stress adaptation in neonates.

Our study has strengths and some limitations. One of its key strengths lies in the new investigation of *SLC6A4* methylation in full-term and preterm infants, utilizing advanced genetic sequencing techniques, which contributes valuable insights into the epigenetic mechanisms underlying early development. Additionally, the study cohorts were created to investigate the course of methylation in multiple genes, tracking participants over time, and collecting data at three distinct points within the first month of life. This approach enabled the analysis of temporal dynamics in *SLC6A4* methylation, providing valuable information on epigenetic changes in preterm and term neonates. However, the study faced challenges, including a relatively small sample size, constrained by logistical and financial considerations. The exploratory nature of current studies prevented accurate sample size calculations and further limited the determination of statistical power. Additionally, a lack of methylation data for full-term infants at D30 limited comparative analysis at these critical time points. However, the use of mixed-effect models allowed for the prediction of methylation at D30 in full-term infants. Other significant limitations of our study included the failure to assess the *SLC6A4* polymorphism pattern in our cohort, the absence of direct, validated measures to quantify the degree of neonatal stress or pain exposure, and the lack of correlation between the epigenetic findings and their biological significance. Although we considered several clinical indicators related to neonatal intensive care, such as the number of skin-breaking procedures, respiratory support, and other interventions, as proxies for early-life stress, these indirect measures cannot replace standardized tools such as the Neonatal Infant Stressor Scale (NISS). Future research in this domain should incorporate structured and validated assessments of neonatal stress and pain to better characterize their potential impact on methylation patterns, which could, in turn, shed light on the long-term implications of *SLC6A4* methylation in preterm infants and its association with brain development, mental health, and broader health outcomes later in life.

## 5. Conclusions

The neurodevelopmental outcomes of preterm children are shaped by complex mechanisms, with early adverse experiences and epigenetic modifications in stress-related genes playing a critical role. This study examined *SLC6A4* promoter methylation in preterm and full-term infants during the first month of life, revealing CpG site–specific and temporal differences between groups. Longitudinal mixed-effects analysis showed that full-term infants exhibited broader CpG-specific increases during early postnatal life, while very and extremely preterm infants displayed more restricted changes, reflecting divergent epigenetic trajectories and potentially distinct pathways of neurodevelopmental adaptation. These findings underscore the dynamic regulation of *SLC6A4* in relation to gestational maturity and early-life stress and emphasize the need for larger, diverse cohorts to clarify how environmental factors shape the epigenome and neurodevelopment. Future studies should (i) correlate *SLC6A4* promoter methylation with gene expression in these infants; (ii) investigate downstream biological pathways affected by altered SLC6A4 regulation; and (iii) evaluate the potential of these methylation patterns as biomarkers for neurodevelopmental outcomes or stress responses in preterm infants. Addressing these aims will help translate epigenetic insights into biomarkers and therapeutic targets to improve preterm infant health.

## Figures and Tables

**Figure 1 biomedicines-13-02753-f001:**
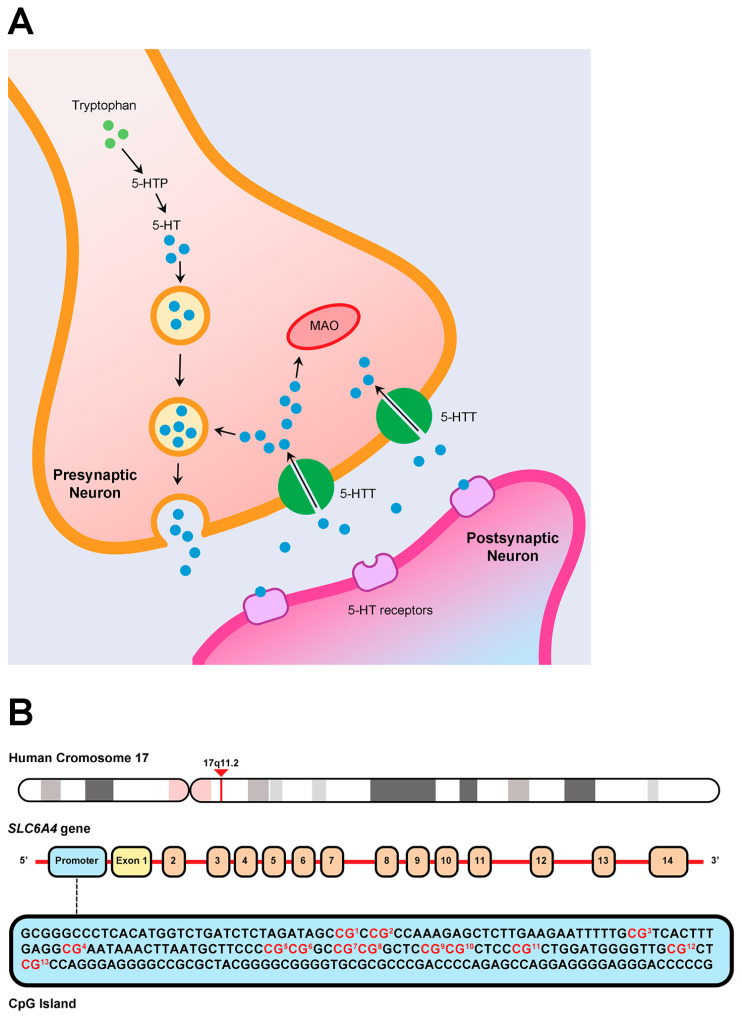
Overview of serotoninergic neurotransmission and the genomic location of *SLC6A4*. (**A**) Serotonin (5-HT) reuptake from the synaptic cleft via serotonin transporters (5-HTT). (**B**) Chromosomal location of the *SLC6A4* gene, highlighting the CpG island located within the promoter region, adjacent to exon 1. The nucleotides highlighted in red correspond to the 13 CpG sites analyzed in this study. (5-HTP = 5-hydroxytryptophan; MAO = monoamine oxidase).

**Figure 2 biomedicines-13-02753-f002:**
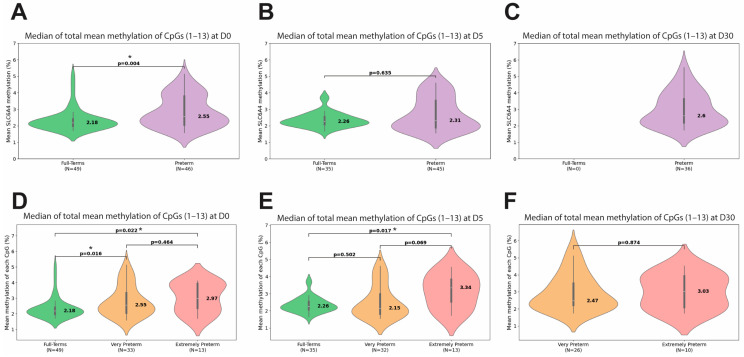
Average methylation percentage across all 13 CpG sites analyzed at three time points: D0 (birth), D5 (fifth day), D30 (thirtieth day). Unstratified preterm group (**A**–**C**). Stratified preterm group into very preterm and extremely preterm (**D**–**F**). The asterisk (*) indicates *p*-values that remained significant after adjustment using the Benjamini–Hochberg false discovery rate (FDR) method.

**Figure 3 biomedicines-13-02753-f003:**
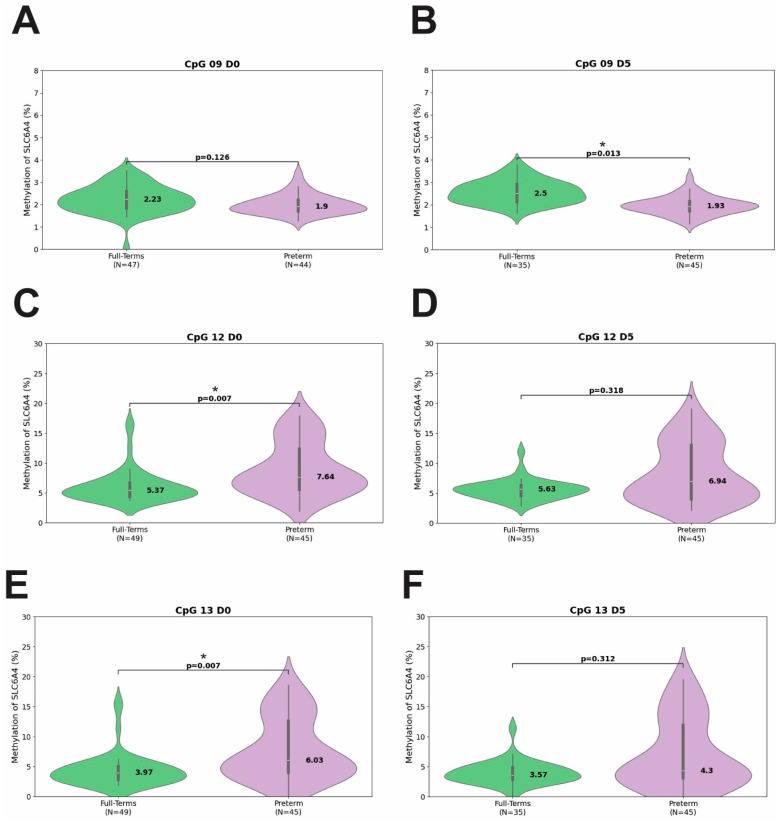
Methylation percentages at CpGs 9 (**A**,**B**), 12 (**C**,**D**), and 13 (**E**,**F**) in term and preterm newborns. Data are at two time points: D0 (birth) and D5 (fifth day). The asterisk (*) indicates *p*-values that remained significant after adjustment using the Benjamini–Hochberg false discovery rate (FDR) method.

**Figure 4 biomedicines-13-02753-f004:**
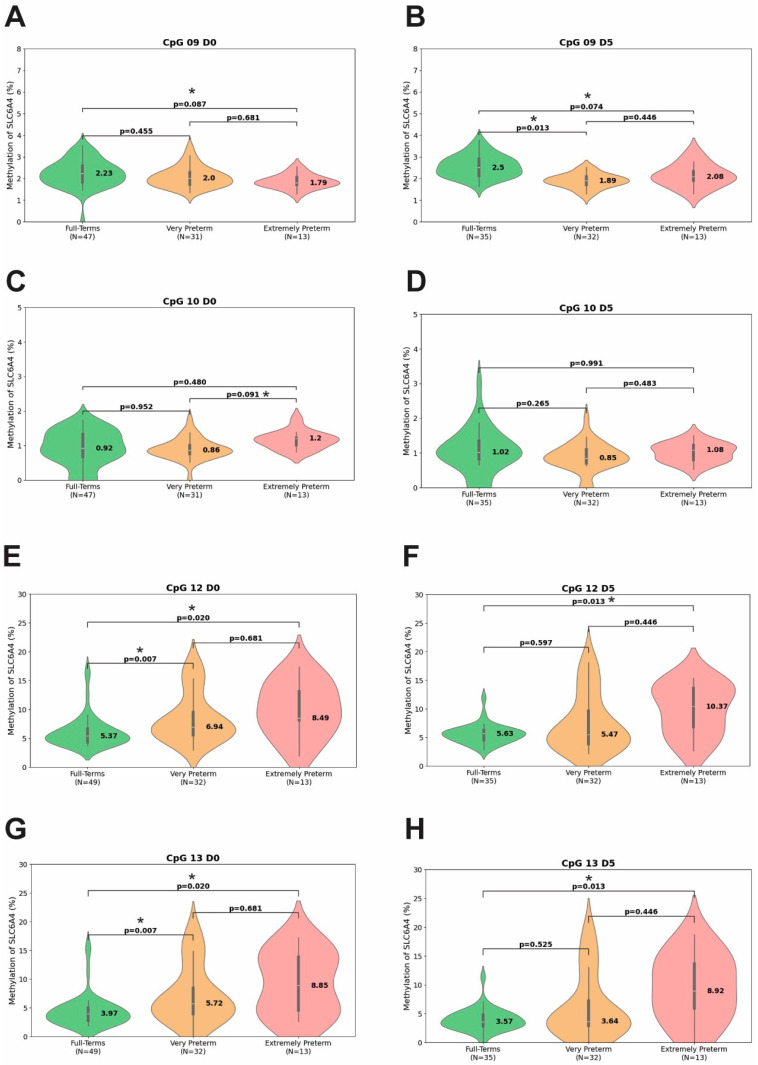
Methylation percentages at CpGs 9 (**A**,**B**), 10 (**C**,**D**), 12 (**E**,**F**), and 13 (**G**,**H**) in term, very preterm, and extremely preterm newborns. Data are for two time points: D0 (birth) and D5 (fifth day). The asterisk (*) indicates *p*-values that remained significant after adjustment using the Benjamini–Hochberg false discovery rate (FDR) method.

**Figure 5 biomedicines-13-02753-f005:**
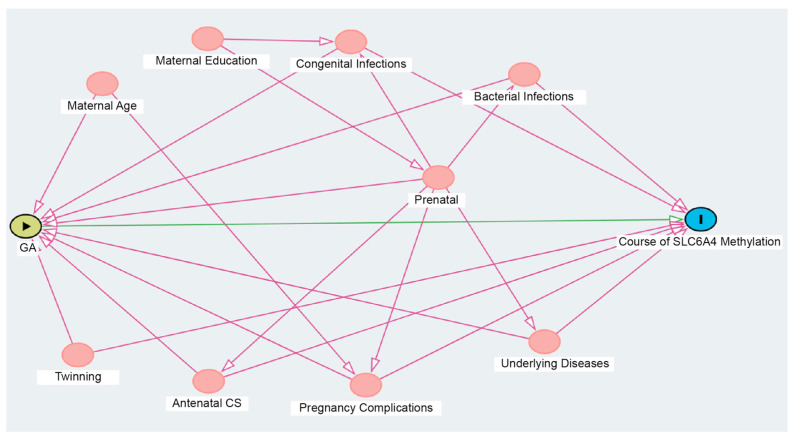
Directed Acyclic Graph (DAG) representing the leading causal association of interest between gestational age, as an “exposure” (preterm vs. term), and the course of *SLC6A4* methylation, as an “outcome”, as well as the possible confounding variables. GA: gestational age; Prenatal: prenatal care; Underlying Diseases: maternal pre-existing comorbidities; Antenatal CS: antenatal corticosteroids.

**Figure 6 biomedicines-13-02753-f006:**
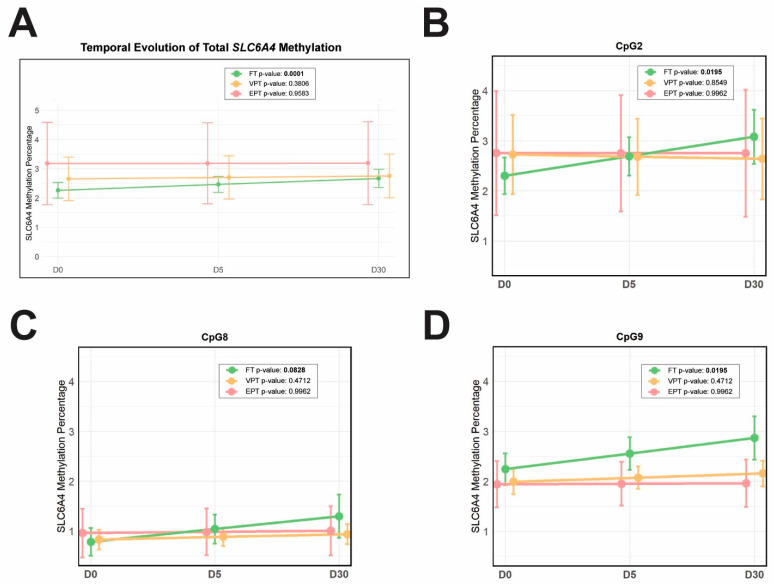
Longitudinal analysis of *SLC6A4* methylation in full-term (FT, green), very preterm (VPT, yellow), and extremely preterm (EPT, pink) infants at D0, D5, and D30, showing the estimated time course of methylation percentages. Points represent the estimated mean methylation (%) from the longitudinal mixed-effects model, with vertical bars indicating 95% confidence intervals, accounting for repeated measures and individual variability. (**A**) Overall methylation (mean of the 13 CpG sites) exhibiting a significant increase in *SLC6A4* methylation in FT during the first month of life. (**B**–**D**) CpGs 2, 8, and 9, showing significant increases in FT infants. Only CpG sites reaching statistical significance are shown. Significant *p*-values, adjusted using the Benjamini–Hochberg false discovery rate (FDR) method, are shown in bold within the box to the upper right of each plot.

**Table 1 biomedicines-13-02753-t001:** Primers used for PCR and pyrosequencing.

Primer	Sequence (5′-3′) ^a^	Sequence to Analyze ^b^	CpG Sites	Product Size
Forward	GGTTTTTATATGGTTTGATTTTTAGA	T**Y**GT**Y**GTTAA AGAGTTTTTG AAGAATTTTT G	1–2	70 bp
Reverse	/5Biosg/CAAAATAACCCAAAAATTCTTCAAAAACT			
Sequencing	TTTATATGGTTTGATTTTTAGATAG			
Forward	ATATGGTTTGATTTTTAGATAGTAGT	TTTTG**Y**GTTA TTTTGAGG**Y**G AATAAATTTA ATGTTTTTT **Y**G**Y**GGT**Y**G**Y**G GTTT**Y**G**Y**GTT TT**Y**GTT	3–11	128 bp
Reverse	/5Biosg/AACCCAACCCCATCCAAC			
Sequencing	GTTAAAGAGTTTTTGAAGAAT			
Forward	TGAGGCG4AATAAATTTAATGTT	TTG**Y**GTT**Y**GT TAGGGAGGGG TYGYGTTAYG GGGYGGGGTG YGYGTTYGAT TTTAGA	12–13	132 bp
Reverse	/5Biosg/CCCCTCCTAACTCTAAAAT			
Sequencing	GTTTTAGTTGGATGGGG			

^a^ Primers designed to target the first 13 CpG sites of the region 28562750-28562958 of chromosome 11 (GRCh37). ^b^ CpG sites analyzed displayed as **Y**.

**Table 2 biomedicines-13-02753-t002:** Epidemiological and clinical characteristics of pregnant women, preterm, and term newborns.

Characteristics	Preterm (*n* = 46)	Full-Term (*n* = 49)
**Maternal Age (years): median (IQR)**	31 (25.2–34)	27 (23.5–33.5)
**Maternal Education (years): *n* (%)**		
1–4	1 (2.2)	0 (0)
5–8	7 (15.2)	9 (18.4)
9–11	8 (17.4)	13 (26.5)
>12	30 (65.2)	27 (55.1)
**^a^ Family Income (national minimum wage): *n* (%)**		
<1	13 (28.3)	4 (8.2)
1–2	17 (37)	28 (57.1)
>2	16 (34.8)	15 (30.6)
**Ethnicity: *n* (%)**		
White	19 (41.3)	22 (44.9)
Black	12 (26.1)	5 (10.2)
Brown-skinned (mixed-race ancestry)	15 (32.6)	22 (44.9)
**Health Conditions**		
**Smokers: *n* (%)**		
No	43 (93.5)	46 (93.9)
Yes	3 (6.5)	3 (6.1)
**Alcohol Consumption: *n* (%)**		
No	42 (91.3)	48 (98)
Yes	4 (8.7)	1 (2)
**Perinatal Consultations: median (IQR)**	5 (4–7)	9 (7–10)
**Gestational Age (weeks): median (IQR)**	28 (27–30)	39 (38–40)
**Apgar Score (1 min): median (IQR)**	7 (4.2–8)	8 (8–9)
**Apgar Score (5 min): median (IQR)**	8.5 (8–9)	9 (9–9)
**Sex: *n* (%)**		
Male	25 (54.3)	25 (51)
Female	20 (43.5)	24 (49)
Undetermined	1 (2.2)	0 (0)
**Birth Weight (g)**		
Mean (SD)	1074.2 (289.9)	3393.6 (385.0)
Median (IQR)	1075 (860–1345)	3320 (3150–3575)
**Length (cm)**		
Mean (SD)	35.9 (1.8)	48.9 (3.4)
Median (IQR)	36.35 (34–37.5)	49 (47.5–49.6)
**Head Circumference (cm)**		
Mean (SD)	26 (2.4)	34.4 (1.5)
Median (IQR)	26 (24.7–28)	34 (33.5–35)

^a^ BRL: Brazilian Real (Currency); IQR: Interquartile range; SD: Standard deviation.

**Table 3 biomedicines-13-02753-t003:** Comparison of median methylation percentages between term and preterm newborns at different time points.

	CpG	Full-Term (%) (IQR)	Preterm (%) (IQR)	*p*-Value ^a^
**D0** Full-term (*n* = 49) Preterm (*n* = 46)	1	2.28 (1.76–2.81)	2.36 (1.62–2.85)	0.983
	2	2.14 (1.70–2.86)	2.47 (1.97–3.29)	0.263
	3	1.96 (1.74–2.35)	2.14 (1.70–2.65)	0.953
	4	2.16 (1.70–2.41)	2.08 (1.85–2.40)	0.983
	5	2.57 (2.18–2.93)	2.65 (2.32–3.08)	0.953
	6	1.01 (0.84–1.27)	1.11 (0.89–1.26)	0.709
	7	1.32 (1.00–1.59)	1.37 (1.10–1.66)	0.709
	8	0.81 (0.57–1.14)	0.86 (0.71–1.09)	0.953
	9	2.23 (1.85–2.58)	1.91 (1.71–2.20)	0.126
	10	0.92 (0.68–1.33)	0.92 (0.81–1.20)	0.983
	11	1.46 (1.03–1.95)	1.44 (1.14–1.84)	0.983
	12	5.37 (4.39–6.64)	7.64 (5.58–12.35)	**0.007**
	13	3.97 (2.78–5.01)	6.03 (4.01–12.59)	**0.007**
**D5** Full-term (*n* = 35) ^b^ Preterm (*n* = 45) ^b^	1	2.38 (1.94–3.07)	2.09 (1.71–3.24)	0.594
	2	2.48 (2.16–2.86)	2.05 (1.84–3.06)	0.318
	3	2.23 (1.84–2.32)	2.07 (1.74–2.44)	0.594
	4	2.11 (1.84–2.46)	2.01 (1.75–2.22)	0.318
	5	2.74 (2.41–3.05)	2.72 (2.29–3.01)	0.594
	6	1.12 (0.95–1.35)	1.1 (0.85–1.33)	0.594
	7	1.49 (1.25–1.76)	1.34 (1.12–1.59)	0.318
	8	1.07 (0.71–1.24)	0.88 (0.71–1.14)	0.318
	9	2.5 (2.14–2.91)	1.93 (1.72–2.14)	**0.013**
	10	1.02 (0.83–1.34)	0.88 (0.74–1.17)	0.318
	11	1.57 (1.28–2.21)	1.39 (1.21–1.73)	0.318
	12	5.63 (4.6–6.27)	6.94 (4.01–12.98)	0.318
	13	3.57 (2.85–4.83)	4.3 (3.11–11.9)	0.312
**D30** Full-term (*n* = 0) ^b^ Preterm (*n* = 36) ^b^	1	N/A	2.26 (1.77–2.75)	N/A
	2	N/A	2.60 (1.85–3.04)	N/A
	3	N/A	2.28 (1.93–2.51)	N/A
	4	N/A	2.18 (1.98–2.64)	N/A
	5	N/A	2.95 (2.57–3.22)	N/A
	6	N/A	1.20 (1.04–1.48)	N/A
	7	N/A	1.40 (1.21–1.57)	N/A
	8	N/A	0.98 (0.78–1.17)	N/A
	9	N/A	2.10 (1.91–2.37)	N/A
	10	N/A	1.06 (0.92–1.26)	N/A
	11	N/A	1.53 (1.32–1.80)	N/A
	12	N/A	7.21 (5.55–13.69)	N/A
	13	N/A	6.24 (3.97–13.92)	N/A

^a^ Mann–Whitney U test. In bold, significant *p*-values adjusted by the Benjamin–Hochberg method (False Discovery Rate—FDR); ^b^ The difference to the initial “*n*” represents samples lost to follow-up. D0: Day of birth; D5: Fifth day of life; D30: Thirtieth day of life; IQR: Interquartile range; N/A: Not available.

**Table 4 biomedicines-13-02753-t004:** Comparison of SLC6A4 methylation among full-term, very preterm ^a^ and extremely preterm ^b^ newborns at different time points.

	CpG	Full-Term (%) (IQR)	Very Preterm (%) (IQR)	Extremely Preterm (%) (IQR)	FT vs. VPT *p*-Value ^c^	FT vs. EPT *p*-Value ^c^	VPT vs. EPT *p*-Value ^c^
**D0** Full-term (*n* = 49) Very preterm (*n* = 33) Extremely preterm (*n* = 13)	1	2.28 (1.76–2.81)	2.40 (1.61–3.25)	2.23 (1.71–2.63)	0.952	0.749	0.681
	2	2.14 (1.70–2.86)	2.51 (1.96–3.36)	2.42 (2.03–2.90)	0.394	0.556	0.915
	3	1.96 (1.74–2.35)	2.07 (1.61–2.52)	2.32 (1.78–2.67)	1.000	0.479	0.681
	4	2.16 (1.70–2.41)	2.07 (1.85–2.30)	2.24 (1.91–2.48)	0.952	0.629	0.681
	5	2.57 (2.18–2.93)	2.57 (2.32–3.08)	2.74 (2.44–3.13)	1.000	0.602	0.681
	6	1.01 (0.84–1.27)	1.05 (0.88–1.22)	1.20 (0.99–1.55)	0.952	0.358	0.681
	7	1.32 (1.00–1.59)	1.35 (1.15–1.50)	1.54 (1.02–1.85)	0.952	0.479	0.681
	8	0.81 (0.57–1.14)	0.84 (0.69–1.05)	0.88 (0.80–1.12)	0.952	0.582	0.681
	9	2.23 (1.85–2.58)	2.00 (1.72–2.29)	1.79 (1.71–2.05)	0.455	**0.087**	0.681
	10	0.92 (0.68–1.33)	0.86 (0.77–1.01)	1.20 (1.02–1.22)	0.952	0.479	**0.091**
	11	1.46 (1.03–1.95)	1.51 (1.14–1.81)	1.27 (1.18–1.89)	1.000	0.932	1.000
	12	5.37 (4.39–6.64)	6.94 (5.58–9.50)	8.49 (8.20–13.13)	**0.007**	**0.020**	0.681
	13	3.97 (2.78–5.01)	5.73 (3.95–8.44)	8.85 (4.55–13.91)	**0.007**	**0.020**	0.681
**D5** Full-term (*n* = 35) Very preterm (*n* = 32) Extremely preterm (*n* = 13)	1	2.38 (1.94–3.07)	2.02 (1.69–3.03)	2.41 (1.87–3.55)	0.525	0.991	0.609
	2	2.48 (2.16–2.86)	2.07 (1.81–2.51)	2.03 (1.93–3.79)	0.264	0.991	0.598
	3	2.23 (1.84–2.32)	2.08 (1.74–2.31)	1.92 (1.75–2.45)	0.656	0.991	0.950
	4	2.11 (1.84–2.46)	2.02 (1.76–2.18)	1.99 (1.58–2.22)	0.264	0.982	0.950
	5	2.74 (2.41–3.05)	2.56 (2.14–2.93)	2.77 (2.42–3.21)	0.431	0.991	0.445
	6	1.12 (0.95–1.35)	1.07 (0.85–1.33)	1.11 (1.05–1.24)	0.495	0.991	0.598
	7	1.49 (1.25–1.76)	1.29 (1.12–1.58)	1.55 (1.27–1.67)	0.264	0.991	0.483
	8	1.07 (0.71–1.24)	0.82 (0.66–1.14)	0.94 (0.74–1.13)	0.329	0.991	0.598
	9	2.50 (2.14–2.91)	1.89 (1.72–2.08)	2.08 (1.92–2.34)	**0.013**	**0.074**	0.445
	10	1.02 (0.83–1.34)	0.85 (0.71–1.10)	1.08 (0.80–1.23)	0.264	0.991	0.483
	11	1.57 (1.28–2.21)	1.39 (1.17–1.72)	1.49 (1.30–1.75)	0.264	0.991	0.483
	12	5.63 (4.60–6.27)	5.47 (3.87–9.67)	10.37 (6.84–13.65)	0.597	**0.013**	0.445
	13	3.57 (2.85–4.83)	3.64 (2.91–7.21)	8.92 (5.98–13.66)	0.525	**0.013**	0.445
**D30** Full-term (*n* = 0) Very preterm (*n* = 26) Extremely preterm (*n* = 10)	1	N/A	2.57 (1.96–2.94)	2.11 (1.75–2.60)	N/A	N/A	0.965
	2	N/A	2.70 (2.03–3.20)	2.47 (1.83–2.67)	N/A	N/A	0.967
	3	N/A	2.23 (1.97–2.52)	2.34 (1.92–2.45)	N/A	N/A	0.967
	4	N/A	2.18 (2.03–2.57)	2.30 (1.88–2.81)	N/A	N/A	0.967
	5	N/A	3.08 (2.83–3.29)	2.57 (2.39–2.83)	N/A	N/A	0.104
	6	N/A	1.20 (1.06–1.49)	1.16 (1.02–1.43)	N/A	N/A	0.967
	7	N/A	1.45 (1.18–1.63)	1.36 (1.25–1.48)	N/A	N/A	0.967
	8	N/A	0.95 (0.77–1.10)	1.16 (0.90–1.27)	N/A	N/A	0.936
	9	N/A	2.22 (2.02–2.38)	1.90 (1.81–2.02)	N/A	N/A	0.104
	10	N/A	1.06 (0.83–1.27)	1.05 (0.96–1.22)	N/A	N/A	0.972
	11	N/A	1.57 (1.29–1.80)	1.40 (1.37–1.77)	N/A	N/A	0.972
	12	N/A	6.83 (5.58–13.06)	9.41 (5.56–14.31)	N/A	N/A	0.967
	13	N/A	5.74 (3.93–13.06)	8.24 (4.56–14.46)	N/A	N/A	0.967

^a^ Very preterm: 28 to less than 32 weeks gestational age; ^b^ Extremely preterm: less than 28 weeks gestational age; ^c^ Mann–Whitney U test. In bold, significant *p*-values. D0: Day of birth; D5: Fifth day of life; D30: Thirtieth day of life; IQR: Interquartile range; FT: Full-term; VPT: Very preterm; EPT: Extremely preterm; N/A: Not available.

**Table 5 biomedicines-13-02753-t005:** Results of the mixed-effect models for change over time in each group, adjusted by antenatal corticosteroid, gestational complications, maternal underlying diseases, twinning, neonatal bacterial infection and congenital infection: predicted methylation values on day of birth (D0), fifth day of life (D5) and thirtieth day of life (D30) and beta coefficient for change over time in each group. * *p*-values adjusted by the BH method (False Discovery rate—FDR)—continued on the next page.

CpG	Day	Full-Term				Very Preterm				Extremely Preterm			
		Predicted Values	Beta	EP	*p*-Value *	Predicted Values	Beta	EP	*p*-Value *	Predicted Values	Beta	EP	*p*-Value *
	D0	2.2840	0.2502	0.1255	0.1407	2.5276	0.0226	0.1212	0.8707	2.4159			
**CpG1**	D5	2.5342				2.5502				2.4147	−0.0012	0.2485	0.9962
	D30	2.7844				2.5729				2.4135			
	D0	2.2988	0.3901	0.1221	**0.0195**	2.7257	−0.0437	0.1228	0.8549	2.7549			
**CpG2**	D5	2.6889				2.6820				2.7531	−0.0017	0.2236	0.9962
	D30	3.0790				2.6383				2.7514			
	D0	2.0700	0.0102	0.1010	0.9202	2.0492	0.0742	0.0639	0.4712	2.2538			
**CpG3**	D5	2.0802				2.1233				2.1974	−0.0565	0.0839	0.9962
	D30	2.0904				2.1409				2.1409			
	D0	2.1461	0.1095	0.0949	0.4167	2.7236	0.0756	0.0655	0.4712	2.1242			
**CpG4**	D5	2.2557				2.7991				2.2067	0.0825	0.1043	0.9962
	D30	2.3652				2.8747				2.2892			
	D0	2.7022	0.1615	0.1628	0.4563	2.6272	0.1802	0.0676	0.1326	2.8331			
**CpG5**	D5	2.8637				2.8074				2.8410	0.0079	0.1271	0.9962
	D30	3.0252				2.9877				2.8488			
	D0	0.9977	0.1300	0.1014	0.4082	1.0382	0.0744	0.0398	0.4349	1.2142			
**CpG6**	D5	1.1277				1.1126				1.2288	0.0146	0.0699	0.9962
	D30	1.2577				1.1870				1.2434			
	D0	1.1674	0.2646	0.1226	0.1229	1.3246	0.0190	0.0481	0.8549	1.5331			
**CpG7**	D5	1.4320				1.3436				1.4474	−0.0857	0.0787	0.9962
	D30	1.6966				1.3628				1.3617			
	D0	0.7848	0.2572	0.1046	**0.0828**	0.8312	0.0537	0.0403	0.4712	0.9599			
**CpG8**	D5	1.0421				0.8850				0.9831	0.0232	0.0694	0.9962
	D30	1.2993				0.9387				1.0062			
	D0	2.2437	0.3118	0.0903	**0.0195**	1.9935	0.0813	0.0565	0.4712	1.9423			
**CpG9**	D5	2.5555				2.0749				1.9525	0.0102	0.0796	0.9962
	D30	2.8673				2.1562				1.9626			
	D0	0.9411	0.0702	0.1241	0.6796	0.8680	0.0673	0.0518	0.4712	1.1313			
**CpG10**	D5	1.0113				0.9353				1.0688	−0.0626	0.0413	0.9962
	D30	1.0815				1.0027				1.0062			
	D0	1.4758	0.2669	0.2067	0.4082	1.4357	0.0362	0.0576	0.7680	1.6116			
**CpG11**	D5	1.7427				1.4719				1.5489	−0.0627	0.0748	0.9962
	D30	2.0096				1.4863				1.4863			
	D0	6.0658	0.0852	0.2424	0.7879	7.9115	−0.0367	0.2241	0.8707	10.5423			
**CpG12**	D5	6.1510				7.8748				10.7397	0.1974	0.6119	0.9962
	D30	6.2362				7.8381				10.9770			
	D0	4.3856	0.2136	0.2259	0.4563	6.4420	0.2127	0.3016	0.7680	9.9944			
**CpG13**	D5	4.5992				6.6547				10.1319	0.1375	0.6956	0.9962
	D30	4.8128				6.8674				10.2693			
Total average methylation	D0	2.2632	0.2033	0.0471	**0.0001**	2.6559	0.0491	0.0555	0.3806	3.1807			
	D5	2.4664				2.7050				3.1875	0.0068	0.1295	0.9583
	D30	2.6697				2.7541				3.1944			

## Data Availability

The data presented in this study are available upon request from the corresponding author due to privacy and ethical reasons.

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
