# Peer review of "Prematurity and Epigenetic Regulation of *SLC6A4*: Longitudinal Insights from Birth to the First Month of Life"

_biomedicines, 2025, doi:10.3390/biomedicines13112753_

Round 1

Reviewer 1 Report

Comments and Suggestions for Authors

This study investigates DNA methylation levels at 13 CpG sites in the promoter region of the Solute 54 Carrier Family 6 Member 4 (SLC6A4) gene in preterm and full-term infants at three time points (D0, D5, and D30). The authors report CpG site-specific and temporal differences in methylation between the two groups, suggesting divergent epigenetic trajectories for neurodevelopment and stress adaptation in neonates. However, several major issues require clarification.

  1. The analysis of DNA methylation at 13 CpG sites in the SLC6A4 promoter suggests that gestational maturity influences methylation patterns. The authors report that full-term infants exhibit greater epigenetic plasticity over time, while preterm infants display more stable signatures.  However, the biological significance of these changes is not explored. Key unresolved questions are whether these methylation dynamics regulate SLC6A4 gene expression, what downstream biological pathways are affected, and whether they hold potential as clinically useful biomarkers.
  2. The abstract lacks a brief introduction to the SLC6A4 It is recommended to state its full name (Solute Carrier Family 6 Member 4) and its primary function to establish the biological context and importance of the study for a broad audience.
  3.  

Other minor issues:

  1. The axis labels and data point fonts in all figures are too small to be read clearly.
  2. Please adjust the formatting to ensure Table 2 appears on a single page.
  3. Table 5 is not displaying completely; several headers and data entries are truncated and need to be adjusted for full visibility.

Author Response

Comments 1

This study investigates DNA methylation levels at 13 CpG sites in the promoter region of the Solute 54 Carrier Family 6 Member 4 (SLC6A4) gene in preterm and full-term infants at three time points (D0, D5, and D30). The authors report CpG site-specific and temporal differences in methylation between the two groups, suggesting divergent epigenetic trajectories for neurodevelopment and stress adaptation in neonates. However, several major issues require clarification.

  1. The analysis of DNA methylation at 13 CpG sites in the SLC6A4promoter suggests that gestational maturity influences methylation patterns. The authors report that full-term infants exhibit greater epigenetic plasticity over time, while preterm infants display more stable signatures. However, the biological significance of these changes is not explored. Key unresolved questions are whether these methylation dynamics regulate SLC6A4 gene expression, what downstream biological pathways are affected, and whether they hold potential as clinically useful biomarkers.

Response 1

We thank the reviewer for raising this critical point. In fact, the objective of our study was to describe the methylation pattern of the SLC6A4 promoter region in preterm newborns compared with term newborns at three time points from birth to the thirtieth day of life. However, we did not intend to describe the possible physiological significance of the patterns found. Indeed, we believe our contribution was the description of distinct SLC6A4 methylation patterns associated with gestational maturity. Future research should prioritize the three main aspects highlighted by the reviewer. In any case, we appreciate the comment, which led us to refine the discussion, adding limitations, changing the conclusions, and outlining directions for subsequent studies.

We have included the following text in the second paragraph of the Discussion section:

“The expression of the SLC6A4 gene, which encodes the serotonin transporter, is influenced by both the genetic polymorphism (5-HTTLPR/rs25531/rs25532) and the methylation state of its promoter region, which is modulated by environmental factors, especially stressful situations. Studies associate the variants "short" and “long_G”,  and the hypermethylation with low gene transcription and, conversely, the “long_A” variant and the hypomethylation with higher transcription (Lesch et al., 1996; Bradley et al., 2005; Zhao et al., 2013; Wendland et al., 2006; Iurescia et al., 2017). The result is a variation in serotonin levels in the synaptic cleft of serotonergic neurons, which may explain human behaviors associated with anxiety and depression (Murthy et al., 2025; Suktas et al., 2024). In preterm infants, who are particularly vulnerable to early stress, understanding SLC6A4 methylation is essential [36,37]. Recent studies suggest that premature birth, in itself, is not necessarily associated with changes in SLC6A4 gene methylation at birth [8,36]. However, these findings are still incipient, and this study aims to contribute to a better understanding of this research area.”

In the paragraph about the study's limitations, we added the following excerpt:

“Other significant limitations of our study included the failure to assess the SLC6A4 polymorphism pattern in our cohort, the absence of direct, validated measures to quantify the degree of neonatal stress or pain exposure, and the lack of correlation between the epigenetic findings and their biological significance. Although we considered several clinical indicators related to neonatal intensive care, such as the number of skin-breaking procedures, respiratory support, and other interventions, as proxies for early-life stress, these indirect measures cannot replace standardized tools such as the Neonatal Infant Stressor Scale (NISS). Future research in this domain should incorporate structured and validated assessments of neonatal stress and pain to better characterize their potential impact on methylation patterns, which could, in turn, shed light on the long-term im-plications of SLC6A4 methylation in preterm infants and its association with brain development, mental health, and broader health outcomes later in life.”

We also decided to revise the study's Conclusions to address the reviewer's suggestions better. The new text is as follows:

“The neurodevelopmental outcomes of preterm children are shaped by complex mechanisms, with early adverse experiences playing a critical role. This study examined SLC6A4 promoter methylation in preterm and full-term infants during the first month of life, revealing CpG site–specific and temporal differences between groups. Longitudinal mixed-effects analysis showed that full-term infants exhibited broader CpG-specific increases during early postnatal life, while very and extremely preterm infants displayed more restricted changes, reflecting divergent epigenetic trajectories and potentially distinct pathways of neurodevelopmental adaptation. These findings underscore the dynamic regulation of SLC6A4 in relation to gestational maturity and early-life stress and emphasize the need for larger, diverse cohorts to clarify how environmental factors shape the epigenome and neurodevelopment. Future studies should: (i) correlate SLC6A4 promoter methylation with gene expression in these infants; (ii) investigate downstream biological pathways affected by altered SLC6A4 regulation; and (iii) evaluate the potential of these methylation patterns as biomarkers for neurodevelopmental outcomes or stress responses in preterm infants. Addressing these aims will help translate epigenetic insights into biomarkers and therapeutic targets to improve preterm infant health.”

References cited in the new excerpts (already present in the reference list of the new manuscript version ): 

Lesch KP, Bengel D, Heils A, Sabol SZ, Greenberg BD, Petri S, Benjamin J, Müller CR, Hamer DH, Murphy DL. Association of anxiety-related traits with a polymorphism in the serotonin transporter gene regulatory region. Science. 1996;274(5292):1527-31. doi: 10.1126/science.274.5292.1527.

Bradley SL, Dodelzon K, Sandhu HK, Philibert RA. Relationship of serotonin transporter gene polymorphisms and haplotypes to mRNA transcription. Am J Med Genet Part B. 2005; 136B(1):58–61. doi: 10.1002/ajmg.b.30185.

Zhao J, Goldberg J, Bremner JD, Vaccarino V. Association between promoter methylation of serotonin transporter gene and depressive symptoms: a monozygotic twin study. Psychosom Med. 2013;75(6):523-9. doi: 10.1097/PSY.0b013e3182924cf4.

Wendland JR, Martin BJ, Kruse MR, Lesch KP, Murphy DL. Simultaneous genotyping of four functional loci of human SLC6A4, with a reappraisal of 5-HTTLPR and rs25531. Mol Psychiatry. 2006;11(3):224-6. doi: 10.1038/sj.mp.4001789.

Iurescia S, Seripa D, Rinaldi M. Looking Beyond the 5-HTTLPR Polymorphism: Genetic and Epigenetic Layers of Regulation Affecting the Serotonin Transporter Gene Expression. Mol Neurobiol. 2017 Dec;54(10):8386-8403. doi: 10.1007/s12035-016-0304-6.

Murthy MK. Molecular pathways linking the serotonin transporters (SERT) to depressive disorder: from mechanisms to treatments. Neuroscience. 2025;584:2-31. doi: 10.1016/j.neuroscience.2025.08.009.

Suktas A, Ekalaksananan T, Aromseree S, Bumrungthai S, Songserm N, Pientong C. Genetic polymorphism involved in major depressive disorder: a systemic review and meta-analysis. BMC Psychiatry. 2024;24(1):716. doi: 10.1186/s12888-024-06195-z.

Comments 2

  1. The abstract lacks a brief introduction to the SLC6A4. It is recommended to state its full name (Solute Carrier Family 6 Member 4) and its primary function to establish the biological context and importance of the study for a broad audience.

Response 2

Thank you for the suggestion. We have added a brief introduction to SLC6A4 in the Abstract, noting its full name (Solute Carrier Family 6 Member 4) and highlighting its primary function as the gene encoding the serotonin transporter, a key component in serotonin reuptake in the synaptic cleft, and its role in the stress response.

Background/Objectives: Prematurity is a significant global health concern, often associated with neurodevelopmental challenges. Solute Carrier Family 6 Member 4 (SLC6A4), the gene encoding the serotonin transporter, a key component in serotonin reuptake in the synaptic cleft, plays a key role in stress response and neurodevelopment. ….

Comments 3

Other minor issues:

  1. The axis labels and data point fonts in all figures are too small to be read clearly.
  2. Please adjust the formatting to ensure Table 2 appears on a single page.
  3. Table 5 is not displaying completely; several headers and data entries are truncated and need to be adjusted for full visibility.

Response 3

We agree and have implemented all requested modifications: we increased the axis labels and data point fonts in all figures for improved readability; adjusted formatting so Table 2 appears on a single page. and fixed Table 5 so all headers and data entries are fully visible. Regarding Table 5, it cannot fit on the same page. For review purposes, we've placed it on two pages. The copy editor can then assist with correcting this issue.

Reviewer 2 Report

Comments and Suggestions for Authors

The manuscript presents an original and well-structured study investigating DNA methylation dynamics in the SLC6A4 gene among full-term, very preterm, and extremely preterm infants at early-life time points. The authors explore how gestational maturity and neonatal stress may influence epigenetic regulation of the serotonin transporter gene, potentially affecting neurodevelopmental outcomes. This topic is highly relevant to neonatal and developmental medicine, given the growing recognition of epigenetic mechanisms linking early adversity to later-life health.

Some comments:

  1. The discussion correctly associates differential methylation with stress-related pathways but could expand on the functional consequences of specific CpG sites (e.g., CpGs 12–13 vs. CpG 9) in transcriptional regulation of SLC6A4. It remains speculative whether the observed methylation changes translate into altered serotonin transporter expression or downstream behavioral outcomes. Please include literature linking specific CpG methylation within the SLC6A4 promoter to gene expression or serotonin levels in relevant tissues.
  2. Neonatal intensive care exposure, medication use (e.g., corticosteroids, sedatives), and maternal factors such as stress, infection, or diet may significantly affect methylation. The manuscript briefly mentions adjusting for several variables but does not specify how neonatal stress exposure was quantified.
    Discuss limitations arising from the absence of direct stress or pain exposure measures (e.g., Neonatal Infant Stressor Scale) and potential confounding by unmeasured variables.

  3. Strengthen conclusions by outlining potential translational implications and future research directions (e.g., linking methylation to neurodevelopmental outcomes at follow-up).

Author Response

Comments to the authors

The manuscript presents an original and well-structured study investigating DNA methylation dynamics in the SLC6A4 gene among full-term, very preterm, and extremely preterm infants at early-life time points. The authors explore how gestational maturity and neonatal stress may influence epigenetic regulation of the serotonin transporter gene, potentially affecting neurodevelopmental outcomes. This topic is highly relevant to neonatal and developmental medicine, given the growing recognition of epigenetic mechanisms linking early adversity to later-life health.

Comments 1

  1. The discussion correctly associates differential methylation with stress-related pathways but could expand on the functional consequences of specific CpG sites (e.g., CpGs 12–13 vs. CpG 9) in transcriptional regulation of SLC6A4. It remains speculative whether the observed methylation changes translate into altered serotonin transporter expression or downstream behavioral outcomes. Please include literature linking specific CpG methylation within the SLC6A4promoter to gene expression or serotonin levels in relevant tissues.

Response 1

We appreciate the comment and agree that differentiation by specific CpG sites is important and deserves further discussion. We have added new paragraphs in the Discussion section (see below), that summarize the literature linking the methylation of individual CpGs in the SLC6A4 promoter region to alterations in transcription, measures of brain serotonin and behavioral phenotypes.

In this new paragraph (below), some sentences were removed and others incorporated to ensure coherence (highlighted in red in the revised version of the manuscript):

“Provenzi et al. (2018) reported recurrent patterns of altered methylation of the SLC6A4 gene after exposure to the neonatal ICU in a systematic review that consolidated findings in preterm infants. This review reported studies that identified specific effects at sites such as CpG2, 5, 6, 16, and 20 across different cohorts, but methodological hetero-geneity (different CpG panels, tissues, and numbering) hinders direct synthesis. Fur-thermore, few studies specified the exact function of individual CpG sites in gene ex-pression in relevant tissues (especially the brain) and in neurobehavioral clinical out-comes, particularly in preterm infants (Provenzi et al., 2018).

In functional model studies, in vitro methylation of a promoter fragment robustly reduces luciferase activity, suggesting a direct mechanism of transcriptional silencing by promoter methylation (Wang et al., 2012). In human samples, Wang et al. (2012) iden-tified a point association between CpG11–12 and reduced 5-HT levels in the orbitofrontal cortex, as measured by PET, supporting the idea that local methylation alterations cor-relate with in vivo serotonin measurements (Wang et al., 2012). Other studies demon-strated a correlation between methylation and reduced levels of SLC6A4 mRNA (Philibert et al., 2008; Vijayendran et al., 2012) and an association between methylation levels in multiple CpG sites in the SLC6A4 promoter region with depressive symptoms, HPA axis reactivity, amygdala activation patterns, anxiety, and burnout (Zhao et al., 2013; Al-exander et al., 2014, Zhu et al., 2023; Vidovic et al, 2023, Bakusic et al., 2021). However, the numbering/index of CpG is not standardized across these studies (each group usually reports the order within the fragment they analyze), which makes a direct correlation between specific sites difficult without mapping the exact genomic distributions. 

In preterm infants, Chau et al. (2014) [42] stated that 7/10 CpG were significantly more methylated in very preterm infants (VPT) of school age (~7 years) (CpG-1, 3–5, 8–10), while CpG-2 showed lower methylation, compared to full-term infants (FT). Meth-ylation increased correlated with behavioral problems, and this effect interacted with neonatal pain exposure and COMT genotype. Although the 10 CpG sites analyzed by Chau et al. differ from those analyzed in this study, they are located within the same CpG island, situated in the promoter region of the SLC6A4 gene, adjacent to exon 1a [42]. In our study, we did not investigate the SLC6A4 polymorphism. The findings of Chau et al. highlight the complex interplay between epigenetics, genetic predisposition, and early stress in shaping neurodevelopmental and behavioral outcomes in preterm children, underscoring the importance of incorporating genotype assessment into epigenetic studies. In a recent longitudinal study, Provenzi et al. (2020) reported that CpG-specific methylation in the neonatal period (a pain-related increase measured at discharge) was a predictor of greater anger expression/greater emotional dysregulation at 4.5 years in preterm infants. Both CpG5 and CpG9 were significantly associated with reactive anger in response to induced stress (Provenzi et al, 2020). As mentioned previously, we used the same numbering system as they do to identify the 13 CpG sites. Therefore, when possible, we recommend either referencing positions to the reference genome or including a comparative table with coordinates to facilitate comparison between studies.

Other studies associate the SLC6A4 gene methylation with neurodevelopmental outcomes without specifying the CpG sites…”

References cited in the new excerpts (already present in the reference list of the new version of the manuscript):

Wang D, Szyf M, Benkelfat C, Provençal N, Turecki G, Caramaschi D, Côté SM, Vitaro F, Tremblay RE, Booij L. Peripheral SLC6A4 DNA methylation is associated with in vivo measures of human brain serotonin synthesis and childhood physical aggression. PLoS One. 2012;7(6):e39501. doi: 10.1371/journal.pone.0039501. Epub 2012 Jun 20. PMID: 22745770; PMCID: PMC3379993.

Zhao J, Goldberg J, Bremner JD, Vaccarino V. Association between promoter methylation of serotonin transporter gene and depressive symptoms: a monozygotic twin study. Psychosom Med. 2013 Jul-Aug;75(6):523-9. doi: 10.1097/PSY.0b013e3182924cf4. Epub 2013 Jun 13. PMID: 23766378; PMCID: PMC3848698.Alexander et al., 2014

Philibert, Robert A., et al. "The relationship of 5HTT (SLC6A4) methylation and genotype on mRNA expression and liability to major depression and alcohol dependence in subjects from the Iowa Adoption Studies." American Journal of Medical Genetics Part B: Neuropsychiatric Genetics 147.5 (2008): 543-549.

Vijayendran, Meeshanthini, et al. "Effects of genotype and child abuse on DNA methylation and gene expression at the serotonin transporter." Frontiers in psychiatry 3 (2012): 55.

Jia-Hui Zhu, Hao-Hui Bo, Bao-Peng Liu, Cun-Xian Jia, The associations between DNA methylation and depression: A systematic review and meta-analysis, Journal of Affective Disorders, 327 (2023):439-450.

Vidovič, E.; Pelikan, S.; Atanasova, M.; Kouter, K.; Pileckyte, I.; Oblak, A.; Novak Šarotar, B.; Videtič Paska, A.; Bon, J. DNA Methylation Patterns in Relation to Acute Severity and Duration of Anxiety and Depression. Curr Issues Mol Biol (2023), 45, 7286–7303.

Bakusic J, Ghosh M, Polli A, Bekaert B, Schaufeli W, Claes S, Godderis L. Role of NR3C1 and SLC6A4 methylation in the HPA axis regulation in burnout. J Affect Disord. 2021 Dec 1;295:505-512. doi: 10.1016/j.jad.2021.08.081. Epub 2021 Aug 31. PMID: 34509065.

Provenzi, L.; Guida, E.; Montirosso, R. Preterm Behavioral Epigenetics: A Systematic Review. Neurosci Biobehav Rev 2018, 84, 262–271.

Comments 2

  1. Neonatal intensive care exposure, medication use (e.g., corticosteroids, sedatives), and maternal factors such as stress, infection, or diet may significantly affect methylation. The manuscript briefly mentions adjusting for several variables but does not specify how neonatal stress exposure was quantified.

Discuss limitations arising from the absence of direct stress or pain exposure measures (e.g., Neonatal Infant Stressor Scale) and potential confounding by unmeasured variables.

Response 2

Indeed, the aim of our study was only to describe and compare the SLC6A4 methylation pattern between preterm and full-term infants. Therefore, we did not incorporate a direct, standardized measure of neonatal stress or pain exposure, such as the Neonatal Infant Stressor Scale (NISS). To address this point, we modified the limitations section to include a commentary about that, as follows:

… “Other significant limitations of our study included the failure to assess the SLC6A4 polymorphism pattern in our cohort, the absence of direct, validated measures to quantify the degree of neonatal stress or pain exposure, and the lack of correlation between the epigenetic findings and their biological significance. Although we considered several clinical indicators related to neonatal intensive care, such as the number of skin-breaking procedures, respiratory support, and other interventions, as proxies for early-life stress, these indirect measures cannot replace standardized tools such as the Neonatal Infant Stressor Scale (NISS). Future research in this domain should incorporate structured and validated assessments of neonatal stress and pain to better characterize their potential impact on methylation patterns, which could, in turn, shed light on the long-term implications of SLC6A4 methylation in preterm infants and its association with brain development, mental health, and broader health outcomes later in life.”

Comments 3

  1. Strengthen conclusions by outlining potential translational implications and future research directions (e.g., linking methylation to neurodevelopmental outcomes at follow-up).

Response 3

Thank you for this valuable suggestion. We have revised the Conclusions section to more clearly outline the potential translational implications of our findings and to specify key future research directions, as recommended. The revised text follows:

“The neurodevelopmental outcomes of preterm children are shaped by complex mechanisms, with early adverse experiences and epigenetic modifications in stress-related genes playing a critical role. This study examined SLC6A4 promoter methylation in preterm and full-term infants during the first month of life, revealing CpG site–specific and temporal differences between groups. Longitudinal mixed-effects analysis showed that full-term infants exhibited broader CpG-specific increases during early postnatal life, while very and extremely preterm infants displayed more restricted changes, reflecting divergent epigenetic trajectories and potentially distinct pathways of neurodevelopmental adaptation. These findings underscore the dynamic regulation of SLC6A4 in relation to gestational maturity and early-life stress and emphasize the need for larger, diverse cohorts to clarify how environmental factors shape the epigenome and neurodevelopment. Future studies should: (i) correlate SLC6A4 promoter methylation with gene expression in these infants; (ii) investigate downstream biological pathways affected by altered SLC6A4 regulation; and (iii) evaluate the potential of these methyl-action patterns as biomarkers for neurodevelopmental outcomes or stress responses in preterm infants. Addressing these aims will help translate epigenetic insights into biomarkers and therapeutic targets to improve preterm infant health.”

Reviewer 3 Report

Comments and Suggestions for Authors

The manuscript entitled "Prematurity and Epigenetic Regulation of SLC6A4: Longitudinal Insights from Birth to the First Month of Life" describes an observational, longitudinal study designed to assess and compare the methylation levels in the promoter region of the SLC6A4 gene between preterm and term neonates at three time points in the first month of life. The study evaluates how prematurity and early-life stress (during NICU hospitalization) might influence variations in the epigenome.

Epigenetic modulation of stress-related genes, such as SLC6A4, has been reported to influence neonatal stress adaptation and developmental outcomes. 

The authors have observed that very preterm and extremely preterm infants displayed more restricted changes, reflecting divergent epigenetic trajectories compared to full-term infants, suggesting reduced epigenetic adaptability, possibly limited stress-response capacity, and potentially distinct pathways of neurodevelopmental adaptation.

The authors apply sound methodological approaches, and the statistical analyses are suitable for addressing the research objectives.

The authors acknowledge the study's small sample size, a lack of methylation data for full-term infants at D30 requiring prediction with mixed-effect models, and the importance of conducting studies with larger and more diverse cohorts to better understand how environmental factors influence the epigenome and neurodevelopment.

I just have a few minor suggestions to improve clarity and reproducibility:

  • Delete lines 91 to 94, it seems like a paragraph from the original journal template hasn’t been properly erased
  • Clarify which confounding variables were included in the model.

Overall, the manuscript is well-structured and clearly written, and I recommend its acceptance after minor revisions.

Author Response

Comments to the authors

The manuscript entitled "Prematurity and Epigenetic Regulation of SLC6A4: Longitudinal Insights from Birth to the First Month of Life" describes an observational, longitudinal study designed to assess and compare the methylation levels in the promoter region of the SLC6A4 gene between preterm and term neonates at three time points in the first month of life. The study evaluates how prematurity and early-life stress (during NICU hospitalization) might influence variations in the epigenome.

Epigenetic modulation of stress-related genes, such as SLC6A4, has been reported to influence neonatal stress adaptation and developmental outcomes. 

The authors have observed that very preterm and extremely preterm infants displayed more restricted changes, reflecting divergent epigenetic trajectories compared to full-term infants, suggesting reduced epigenetic adaptability, possibly limited stress-response capacity, and potentially distinct pathways of neurodevelopmental adaptation.

The authors apply sound methodological approaches, and the statistical analyses are suitable for addressing the research objectives.

The authors acknowledge the study's small sample size, a lack of methylation data for full-term infants at D30 requiring prediction with mixed-effect models, and the importance of conducting studies with larger and more diverse cohorts to better understand how environmental factors influence the epigenome and neurodevelopment.

Comments 1

I just have a few minor suggestions to improve clarity and reproducibility:

  • Delete lines 91 to 94, it seems like a paragraph from the original journal template hasn’t been properly erased.

Response 1

Thank you for pointing that out. It was an oversight. This extra paragraph (lines 91–94) was part of the journal's template and has been removed.

Comments 2

  • Clarify which confounding variables were included in the model. Overall, the manuscript is well-structured and clearly written, and I recommend its acceptance after minor revisions.

Response 2

The confounding variables considered in the longitudinal analysis included: antenatal corticosteroids, gestational complications, maternal underlying diseases, twinning, neonatal bacterial infections, and congenital infections. As indicated in the first paragraph of section 3.3, this minimum set of adjustment variables was obtained by constructing a Directed Acyclic Graph (DAG) using the DAGitty program, available for free online (https://www.dagitty.net/). This is a non-statistical procedure for determining a minimum set of adjustments to prevent overfitting. To clarify, we decided to include a new figure (Figure 5) representing the Directed Acyclic Graph (DAG).

Reviewer 4 Report

Comments and Suggestions for Authors

This is an interesting paper and of significant interest for practitioners who work in neonatal care. 

My main comments are minor. In the Introduction, I would just comment that neonatal practitioners do practice various methods such as pain relief strategies, use of skin to skin care , listening to the human voice and developing early communication strategies with infants on neonatal units to reduce stress and its negative impact on infants. I would mention this again in the Discussion at the end, but stress that these strategies often have small sample number studies to support such methods but that your study actually should provide a motivation for further research to support methods that can help reduce infant stress. 

Author Response

Comments to the authors

This is an interesting paper and of significant interest for practitioners who work in neonatal care. My main comments are minor. In the Introduction, I would just comment that neonatal practitioners do practice various methods such as pain relief strategies, use of skin to skin care, listening to the human voice and developing early communication strategies with infants on neonatal units to reduce stress and its negative impact on infants. I would mention this again in the Discussion at the end, but stress that these strategies often have small sample number studies to support such methods but that your study actually should provide a motivation for further research to support methods that can help reduce infant stress. 

Response

We appreciate the reviewer's thoughtful and constructive feedback on our work. We fully agree with the relevance of current neonatal care practices aimed at reducing stress and its negative impact on infants. We have added a second paragraph in the Introduction, acknowledging these strategies – including pain relief interventions, skin-to-skin care, exposure to the human voice, and early communication approaches – and emphasizing their importance as foundations for improving neonatal outcomes and guiding future research.

On this occasion, we would like to highlight that our group submitted another manuscript to this same Special Edition, reporting our results on the methylation of the SLC6A4 gene in premature newborns undergoing the Kangaroo Mother Care method.

“In neonatal care, several strategies have been implemented to minimize stress and its potential long-term effects, including pain relief interventions, skin-to-skin care (Kangaroo Mother Care), exposure to the human voice, and early communication approaches with infants in the NICU environment (Weaver et al, 2004; Barry et al, 2008; Conradt et al, 2019; Krol et al, 2019; Fontana et al, 2021 and Wigley et al, 2021). Although most of these approaches are supported primarily by animal research and a few human studies with relatively small sample sizes, they represent essential strategies to mitigate the adverse effects of early stress on infant development. Understanding the biological mechanisms underlying these effects, including epigenetic regulation of stress-related genes, may help strengthen the scientific basis for such interventions and guide future clinical strategies.”

References cited in the new excerpts (already present in the reference list of the new version of the manuscript):

Weaver, I.C., Cervoni, N., Champagne, F.A., D'Alessio, A.C., Sharma, S., Seckl, J.R., Dymov, S., Szyf, M., Meaney, M.J. Epigenetic programming by maternal behavior. Nat Neurosci (2004) 7, 847-54.

Barry, R.A.; Kochanska, G.; Philibert, R.A. G x E interaction in the organization of attachment: mothers' responsiveness as a moderator of children's genotypes. J Child Psychol Psychiatry (2008) 49, 1313-20.

Conradt, E.; Ostlund, B.; Guerin, D.; Armstrong, D.A.; Marsit, C.J.; Tronick, E.; LaGasse, L.; Lester, B.M. DNA methylation of NR3c1 in infancy: Associations between maternal caregiving and infant sex. Infant Ment Health J (2019) 40, 513-522.

Krol, K.M.; Moulder, R.G.; Lillard, T.S.; Grossmann, T.; Connelly, J.J. Epigenetic dynamics in infancy and the impact of maternal engagement. Sci Adv (2019) 5, eaay0680.

Fontana, C.; Marasca, F.; Provitera, L.; Mancinelli, S.; Pesenti, N.; Sinha, S.; Passera, S.; Abrignani, S.; Mosca, F.; Lodato, S.; Bodega, B.; Fumagalli, M. Early maternal care restores LINE-1 methylation and enhances neurodevelopment in preterm infants. BMC Med (2021) 19, 42

Mariani Wigley, I.L.C.; Mascheroni, E.; Fontana, C.; Giorda, R.; Morandi, F.; Bonichini, S.; McGlone, F.; Fumagalli, M.; Montirosso, R. The role of maternal touch in the association between SLC6A4 methylation and stress response in very preterm infants. Dev Psychobiol (2021) 63 Suppl 1, e22218. 

Round 2

Reviewer 1 Report

Comments and Suggestions for Authors

 Accept in present form